

# The operational methane retrieval algorithm for TROPOMI

Haili Hu[1], Otto Hasekamp[1], André Butz[2], André Galli[3], Jochen Landgraf[1], Joost Aan de Brugh[1], Tobias Borsdorff[1], Remco Scheepmaker[1], and Ilse Aben[1]

[1]SRON Netherlands Insitute for Space Research, Utrecht, the Netherlands
[2]IMK-ASF, Karlsruhe Institute of Technology, Eggenstein-Leopoldshafen, Germany
[3]Physics Institute, University of Bern, Bern, Switzerland

*Correspondence to:* Haili Hu
(h.hu@sron.nl)

**Abstract.** This work presents the operational methane retrieval algorithm for the Sentinel-5 Precursor (S5-P) satellite and its performance tested on realistic ensembles of simulated measurements. The target product is the column-averaged dry air volume mixing ratio of methane ($XCH_4$), which will be retrieved simultaneously with scattering properties of the atmosphere. The algorithm at-

tempts to fit spectra observed by the shortwave and near-infrared channels of the TROPOMI spectrometer aboard S5-P.

The sensitivity of the retrieval performance to atmospheric scattering properties, atmospheric input data and instrument calibration errors is evaluated. Also, we investigate the effect of inhomogeneous slit illumination on the instrument spectral response function. Finally, we discuss the cloud

filters to be used operationally and as backup.

We show that the required accuracy and precision of $< 1\%$ for the $XCH_4$ product are met for clear sky measurements over land surfaces and after appropriate filtering of difficult scenes. The algorithm is very stable having a convergence rate of 99%. The forward model error is less than 1% for about 95% of the valid retrievals. Model errors in the input profile of water do not influence the

retrieval outcome noticeably. The methane product is expected to meet the requirements if errors in input profiles of pressure and temperature remain below 0.3% and 2 K, respectively. We find further that, of all instrument calibration errors investigated here, our retrievals are the most sensitive to an error in the instrument spectral response function of the short-wave infrared channel.

## 1 Introduction

Methane ($CH_4$) is the most important anthropogenic greenhouse gas after carbon dioxide ($CO_2$). While it occurs in smaller concentrations, it has a higher global warming potential per molecule than $CO_2$. An accurate understanding of $CH_4$ sources and sinks is essential for a reliable prediction of





climate change. Space-based measurements can provide continuous and global monitoring of $CH_4$, leading to much-needed improved constraints on the surface fluxes.

Several current and future satellite missions measure $CH_4$ abundances in the Earth's atmosphere. The SCanning Imaging Absorption spectroMeter for Atmospheric CartograpHY (SCIAMACHY) aboard ENVISAT (Bovensmann et al., 1999)) was the first space-based instrument to measure atmospheric $CH_4$ with sensitivity down to the Earth's surface. Frankenberg et al. (2005) were the first to use these measurements to constrain $CH_4$ surface fluxes. After loss of contact with ENVISAT in
2012, the Greenhouse gases Oberving SATellite (GOSAT) is currently the only satellite measuring atmospheric $CH_4$ (Kuze et al., 2009). While GOSAT has a higher sensitivity and spatial resolution than SCIAMACHY, it has a fairly low spatial sampling. In 2016, the TROPOshpheric Monitoring Instrument (TROPOMI) will be launched aboard the Sentinel-5 Precursor (S5-P) satellite, and it will provide $CH_4$ measurements as one of its key products with unprecedented high precision,
spatial resolution and global daily coverage.

The common goal of the abovementioned missions is to provide atmospheric $CH_4$ concentrations with sufficient accuracy and spatiotemporal coverage to allow the assessment of $CH_4$ sources through inverse modelling. The observation strategy relies on measuring spectra of sunlight, backscattered by the Earth's surface and atmosphere, in the shortwave infrared (SWIR) spectral range. Ab-
sorption features of $CH_4$ molecules allow for retrieval of its atmospheric concentration with high sensitivity down to the Earth's surface where the main $CH_4$ sources are located. The applicability of such measurements for estimating source strengths, however, strongly depends on the precision and accuracy achieved. Residual systematic biases must be well below 1% to facilitate inverse modelling (Meirink et al., 2006; Bergamaschi et al., 2007, 2009).

Scattering by aerosols and cirrus is one of the major challenges for retrievals of $CH_4$ from space-based SWIR observations. While contamination by optically thick clouds can be filtered out reliably, optically thin scatterers are much harder to detect, yet still modify the light path of the observed backscattered sunlight. This can lead to underestimation or overestimation of the true $CH_4$ column if not appropriately accounted for. The net light path effect strongly depends on the amount, size,
and height distribution of the scatterers as well as on the reflectance of the underlying surface (Aben et al., 2007; Gloudemans et al., 2008). Therefore, retrieval strategies rely on inferring the target gas concentration either simultaneously with atmospheric scattering properties or with a light path proxy.

Frankenberg et al. (2005) introduced the "proxy" approach for $CH_4$ retrieval from SCIAMACHY
measurements around 1600 nm, by using the simultaneously retrieved $CO_2$ column as a light path proxy. The proxy approach relies on the assumptions that scattering effects cancel in the ratio of the $CH_4$ column and the $CO_2$ column, and that a prior estimate of the $CO_2$ column is sufficiently accurate to recalculate the $CH_4$ column from the measured $CH_4/CO_2$ ratio. In this case scattering is ignored in the forward modeling. Further applications of the proxy approach for $CH_4$ retrieval from


SCIAMACHY are described by Frankenberg et al. (2008) and Schneising et al. (2011). For GOSAT, the proxy approach has been successfully applied by Parker et al. (2011) and Schepers et al. (2012).

Alternatively, scattering induced light path modification can be taken into account by simultaneously inferring the atmospheric $CH_4$ concentration and physical scattering properties of the atmosphere. Such "physics-based" methods have been developed for space-based $CO_2$ and $CH_4$ measure-

ments from SCIAMACHY, GOSAT, and the Orbiting Carbon Observatory (OCO), see e.g. Connor et al. (2008); Butz et al. (2009); Reuter et al. (2010); O'Dell et al. (2012). The physics-based methods make use of the Oxygen-A band in the near infrared (NIR) around 760 nm and absorption bands of the target absorber in the SWIR spectral range. The advantage of physics-based methods for $CH_4$ retrieval compared to proxy methods is that they do not depend on prior information on the $CO_2$

column. On the other hand, the physics-based algorithms are more complex and computationally expensive. Also, they may be limited by the information content of the measurement with respect to aerosol properties and related forward model errors in the description of aerosols. A detailed comparison between the two methods for GOSAT is provided by Schepers et al. (2012).

TROPOMI has four spectral channels in the ultraviolet (UV), visible (VIS), near infrared (NIR)

and shortwave infrared (SWIR), with spectral ranges of 270–320 nm, 310–495 nm, 675–775 and 2305–2385 nm, respectively. We use TROPOMI measurements in the NIR and SWIR for $CH_4$ retrievals. This spectral range does not allow for a light-path-proxy approach, and thus the effect of aerosols and cirrus needs to be accounted for using a physics-based method as described above. The goal of this paper is to present the $CH_4$ retrieval algorithm for TROPOMI and investigate its

sensitivity to algorithm assumptions, atmospheric input data and instrument calibration errors, and filtering criteria. To this end, we simulated realistic TROPOMI measurements for aerosol and cirrus loaded atmospheres under clear-sky and cloudy conditions.

The outline of this paper is as follows. We start with the methodology in Sect. 2, giving an overview of the instrument, the retrieval algorithm and the methane data product. Then a detailed

error sensivity study is presented in Sect.3, based on methane retrievals on a clear sky global ensemble of simulated spectra. In Sect. 4, a study of cloud filtering is performed on a TROPOMI orbit of simulated spectra that cover a realistic range of cloud parameters. Conclusions are presented in Sect. 5.

## 2   Methodology

The S5-P satellite has a designed 7-year lifetime and will fly in a sun-synchronous orbit at 824 km altitude. Its single payload, TROPOMI, is a push-broom imaging spectrometer with a wide swath of 2600 km and a ground pixel of $7{\times}7$ km$^2$ in exact nadir. Approximately, TROPOMI observes a full swath per second, which results in ∼216 spectra per second. The instrument comprises two spectrometer modules, the first containing the UV, VIS and NIR spectral channels and the second



dedicated to the SWIR channel. We use the NIR and SWIR channels with spectral resolutions of 0.38
and 0.25 nm and spectral sampling ratios of 2.8 and 2.5, respectively (Veefkind et al., 2012). Since
the NIR and SWIR detectors are incorporated in different instrument modules, the NIR spectra will
be coregistered with the SWIR spectra before performing $CH_4$ retrievals. Examples of simulated
TROPOMI NIR and SWIR spectra are shown in Fig. 1.

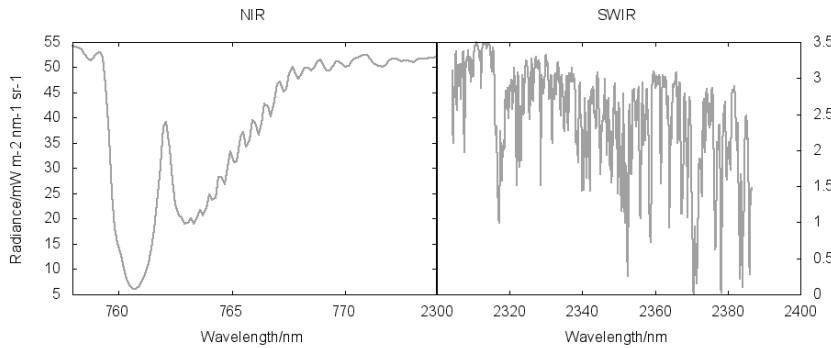

**Fig. 1.** Simulated TROPOMI spectra in the NIR and SWIR channels for a scenario from the global ensemble,
see Sect. 3.1. The scenario, located in a rural area in Utah (USA), is observed at nadir and SZA$= 50°$.

## 2.1 The retrieval algorithm


The S5-P operational $CH_4$ retrieval algorithm is based on RemoTeC, which was orginally developed
for $CO_2$ and $CH_4$ retrievals from OCO and GOSAT obervations (Butz et al., 2009, 2010). A first
performance study on TROPOMI measurements has been done by Butz et al. (2012), where a de-
tailed description of the algorithm can be found. Here, we summarise the essentials and elaborate
on the modifications made since then.

### 2.1.1 The forward model

The algorithm aims at inferring the atmospheric state vector $\mathbf{x}$ from spectral measurements $\mathbf{y}$ in the
NIR (757–774 nm) and SWIR (2305–2385 nm) ranges. This requires a forward model $\mathbf{F}$ that can
accurately compute the measurement given the atmospheric state:

$$\mathbf{y} = \mathbf{F}(\mathbf{x}) + \mathbf{e_y} + \mathbf{e_F}. \tag{1}$$

where $\mathbf{e_y}$ is the measurement noise error, and $\mathbf{e_F}$ is the forward model error. The forward model
incorporates the linearised radiative transfer model LINTRAN (Schepers et al., 2014). LINTRAN
simulates the radiance at the top of the atmosphere $I^{\text{TOA}}$ on a fine internal spectral grid. The
model needs as input a solar irradiance spectrum on the internal spectral grid, which is inferred from



the daily solar measurements of TROPOMI using the deconvolution approach by van Deelen et al.
(2007) and Wassmann et al. (2015). The simulated radiance measurement is obtained by a spectral
convolution of $I^{\mathrm{TOA}}$ with the instrument spectral response function (ISRF).

In addition to accuracy, an important requirement on the algorithm is computational speed. To be
able to process TROPOMI's huge amount of measurements, the CPU time per retrieval has to be in
the order of seconds. Our algorithm achieves this by, among other things, using the linear k-method
for multiple scattering (Hasekamp and Butz, 2008). Single scattering is calculated line-by-line as it
is computationally less expensive.

For a given model atmosphere, the forward model simulates spectra of backscattered sunlight
taking into account absorption and scattering by molecules and particles. The model atmosphere
is defined for 36 pressure-equidistant vertical layers, with the top at 0.1 hPa and bottom at the
surface pressure $p_{\mathrm{surf}}$. We calculate $p_{\mathrm{surf}}$ by interpolating the meteorological surface pressure, from
the European Centre for Medium-Range Weather Forecasts (ECMWF), on the surface elevation,
from a digital elevation map (Danielson and Gesch, 2011; Farr et al., 2007)). The absorbing trace
gases of interest are $O_2$ in the NIR band, and $CH_4$, $H_2O$, and $CO$ in the SWIR band. The first
guess layer subcolumns of these gases are calculated from input profiles of $CH_4$ and $CO$ (from
the global chemical transport model TM5, Houweling et al. (2014)), and temperature, humidity
and pressure profiles (from ECMWF forecast data). Molecular absorption features are calculated
using appropriate spectroscopic databases (Tran et al., 2006; Rothman et al., 2009; Scheepmaker
et al., 2012; Rothman et al., 2013). Here, we evaluate the absorption cross-sections on a 72-layer
equidistant pressure grid to account for the strong temperature- and pressure-dependence of the
cross-sections. Molecular scattering properties are given by Rayleigh theory. Particulate absorption
and scattering are computed with Mie theory using tabulated aerosol properties by Dubovik et al.
(2006).

In our algorithm, the aerosol type is characterised by the refractive index and size distribution. The
complex refractive index is fixed at $1.4 - 0.01i$ in the $O_2$A-band and $1.47 - 0.008i$ in the SWIR band.
The size distribution is described by a power-law function with size parameter $\alpha$ (e.g. Mishchenko
et al. (1999)):

$$n(r) = \begin{cases} A, & \text{if } r \leq r_1. \\ A(r/r_1)^{-\alpha}, & \text{if } r_1 < r \leq r_2. \\ 0, & \text{if } r > r_2. \end{cases} \qquad (2)$$

where $r_1 = 0.1\mu\mathrm{m}$, $r_2 = 10\mu\mathrm{m}$, $r$ is the particle radius, and $A$ is a normalisation constant. The
amount of aerosol and its vertical distribution is provided by the vertically integrated column number
density $N_{\mathrm{aer}}$ and a normalized Gaussian height distribution $h(z_k)$ with $z_k$ the height of layer $k$:

$$h(z_k) = B \exp\left(-\frac{4\ln 2 (z_k - z_{\mathrm{aer}})^2}{w^2}\right)$$





where $w = 2000$ m, $z_{\mathrm{aer}}$ is the central height and B is a normalisation constant. Thus in layer $k$ with thickness $\Delta z_k$, the layer sub-column $n_k$ is given by:

$\quad n_k = N_{\mathrm{aer}} h(z_k) \Delta z_k.$

We have modified the retrieval forward model with respect to Butz et al. (2012) to account for chlorophyll fluorescence emission in a simplified manner as described by Frankenberg et al. (2012). In short, scattering of the fluorescence emission is ignored and solely the spectral shape and absorption features by oxygen ($O_2$) are modeled. This allows fluorescence to be treated as a simple

$\quad$ additive term to the radiance before convolution with the ISRF. The surface emission at the top of the atmosphere (TOA) is then modelled as:

$$F_s(\lambda)^{TOA} = F_{s,755}^{surf}(1 - s(\lambda - 755))e^{-\tau_{O_2}(\lambda)/\mu} \tag{3}$$

where $\tau_{O_2}(\lambda)$ is the vertical optical thickness of oxygen, $\mu$ is the cosine of the viewing zenith angle and $\lambda$ is the wavelength in nanometres.

### 2.1.2 The state vector


The state vector $\mathbf{x}$ consists of 25 elements: a 12-layer vertical profile of $CH_4$ partial column number densities, the total column number densities of $H_2O$ and CO, three scattering parameters $N_{\mathrm{aer}}$, $\alpha$ and $z_{\mathrm{aer}}$ (related to amount, size, and height), the surface albedo (up to first order spectral dependence and in NIR and SWIR band), two terms to account for a spectral shift of the measurement (NIR

$\quad$ and SWIR band) and two terms to account for chlorophyll fluorescence emission, $F_{s,755}^{surf}$ and $s$ (see Eq. (3)). An overview of the state vector elements is given in Table 1.

### 2.1.3 The inversion procedure

The state vector is found by inverting Eq. (1), where the inverse method is based on a Philips-Tikhonov regularization scheme (Phillips, 1962; Tikhonov, 1963). Regularization is required be-

$\quad$ cause the inverse problem is ill-posed, i.e. the measurements $\mathbf{y}$ typically contain insufficient information to retrieve all state vector elements independently. Philips-Tikhonov regularization aims at reducing contributions from measurement noise to the retrieved state vector while retaining valuable information. Because the forward model $\mathbf{F}(\mathbf{x})$ is non-linear in $\mathbf{x}$, the inversion is performed iteratively by a step-size controlled Gauss-Newton scheme, where at each iteration step the forward

$\quad$ model is linearised.

The inverse algorithm finds $\mathbf{x}$ by minimizing the cost function that is the sum of the least-squares cost function and a side constraint weighted by the regularization parameter $\gamma$ according to

$$\hat{\mathbf{x}} = \min_{\mathbf{x}} \left( ||\mathbf{S_y}^{-1/2}(\mathbf{F}(\mathbf{x})) - \mathbf{y})||^2 + \gamma ||\mathbf{W}(\mathbf{x} - \mathbf{x_a})||^2 \right)$$

where $\mathbf{S_y}$ is the diagonal measurement error covariance matrix, which contains the noise estimate.

$\quad \mathbf{x_a}$ is an a priori state vector, and $\mathbf{W}$ is a diagonal weighting matrix that renders the side constraint





**Table 1.** State vector elements of the baseline methane algorithm.

| state vector element |
| --- |
| $CH_4$ sub-columns in 12 vertical layers |
| CO total column |
| $H_2O$ total column |
| aerosol column $N_{aer}$ |
| aerosol size parameter $\alpha$ |
| aerosol height parameter $z_{aer}$ |
| Lambertian surface albedo in NIR band |
| $1^{st}$ order spectral dependence surface albedo in NIR band |
| Lambertian surface albedo in SWIR band |
| $1^{st}$ order spectral dependence surface albedo in SWIR band |
| spectral shift NIR |
| spectral shift SWIR |
| fluorescence emission at 755 nm $F_{s,755}^{surf}$ |
| fluorescence spectral slope $s$ |

dimensionless and ensures that only the $CH_4$ parameters and the scattering parameters contribute to its norm: $W_{jj} = 1/x_{a,j}$ for the $CH_4$ column number densities and the three aerosol parameters, and $W_{jj} = 10^{-7}/x_{a,j}$ for all other state vector elements. The latter are thus retrieved in a least-squares sense. For determining $\gamma$, the L-curve criterion (Hansen, 1998) is applied in the baseline algorithm.

However, $\gamma$ may be finetuned later using real observations.

## 2.2  The $CH_4$ data product

Although we retrieve methane in $n = 12$ sublayers, there is virtually no profile information in the measurement. The degree of freedom of signal of the retrieved methane profile is about 1. Therefore, the methane data product is given as a column-averaged dry air mixing ratio $XCH_4$. This quantity is

obtained from the methane entries of the retrieved state vector $x_i$ through

$$XCH_4 = \sum_{i=1}^{n} x_i / V_{air,dry}$$

where $V_{air,dry}$ is the dry air column (calculated from meteorological input surface pressure and water vapour profile).

To interpret the retrieved $XCH_4$ correctly, one also needs the column averaging kernel $\mathbf{A}_{col}$ that

describes the sensitivity of the retrieved $CH_4$ column to changes to the true methane profile (see





Rodgers (2000) for details):

$$A_{\text{col,i}} = \frac{\partial \sum\limits_{i=1}^{n} x_i}{\partial x_{\text{true},i}}. \tag{4}$$

As an example we show in Fig. 2 the column averaging kernel corresponding to the methane retrieval performed on the simulated spectra of Fig. 1. We see that the column averaging kernel is around 1 in the lower atmosphere. From Eq. (4) it is clear that the closer $\mathbf{A}_{\text{col}}$ is to 1, the more the retrieved column represents the true column (see also Eq. (5)). This illustrates that methane retrievals from the SWIR band have sensitivity down to the ground.

For validation and interpretation purposes it is important to realize that the retrieved $XCH_4$ is related to the true methane profile $\mathbf{x}_{\text{true}}$ and the a priori profile $\mathbf{x}_a$ as:

$$\begin{aligned} XCH_4 = &\sum_{i=1}^{n} \big(A_{\text{col},i}x_{\text{true},i} + (1-A_{\text{col},i})x_{a,i}\big)/V_{\text{air,dry}} \\ &+ \Delta XCH_{4,\mathbf{F}} + \Delta XCH_{4,\mathbf{y}} \end{aligned} \tag{5}$$

where $\Delta XCH_{4,\mathbf{F}}$ is the bias caused by forward model errors and $\Delta XCH_{4,\mathbf{y}}$ is the retrieval noise due to measurement noise. The standard deviation of the retrieval noise, i.e. the precision $\sigma_{XCH_4}$, follows from the error covariance matrix $\mathbf{S}_x$, that describes the effect of measurement noise on the retrieved parameters:

$$\sigma_{XCH_4} = \sqrt{\sum_{i=1}^{n}\sum_{j=1}^{n} S_{x,i,j}}/V_{\text{air,dry}}. \tag{6}$$

Together with $XCH_4$ and $\mathbf{A}_{\text{col}}$, the precision $\sigma_{XCH_4}$ is given in the main data product. Note that for simulations, the bias $\Delta XCH_{4,\mathbf{F}}$ can be calculated from Eq. (5), since all other terms are either known from the simulations or calculated by the retrieval algorithm. In Sect. 3, we evaluate $\Delta XCH_{4,\mathbf{F}}$, $\Delta XCH_{4,\mathbf{y}}$ and their sensitivity to input errors.

### 2.3 Data filtering

Our algorithm has been designed to be effcient and accurate under certain assumptions, e.g. the atmosphere is cloud free and plane-parallel with optically thin scatterers. Therefore it is essential to filter out those scenes for which these assumptions break down to ensure data quality and reach the required accuracy. Moreover, if filtering can be done a priori, for example using cloud detection, we can considerably reduce the number of performed retrievals and hence computation time.

For operational cloud filtering, measurements from the Visible Infrared Imaging Radiometer Suite (VIIRS) aboard the Suomi-NPP satellite will be used. S5-P is foreseen to operate in loose formation with Suomi-NPP, meaning that both missions observe the same ground scene with a time delay of about 5 minutes. In case VIIRS data is not available, we have developed a backup cloud filter using $H_2O$ retrievals from the weak and strong absorption band assuming a non-scattering atmosphere.




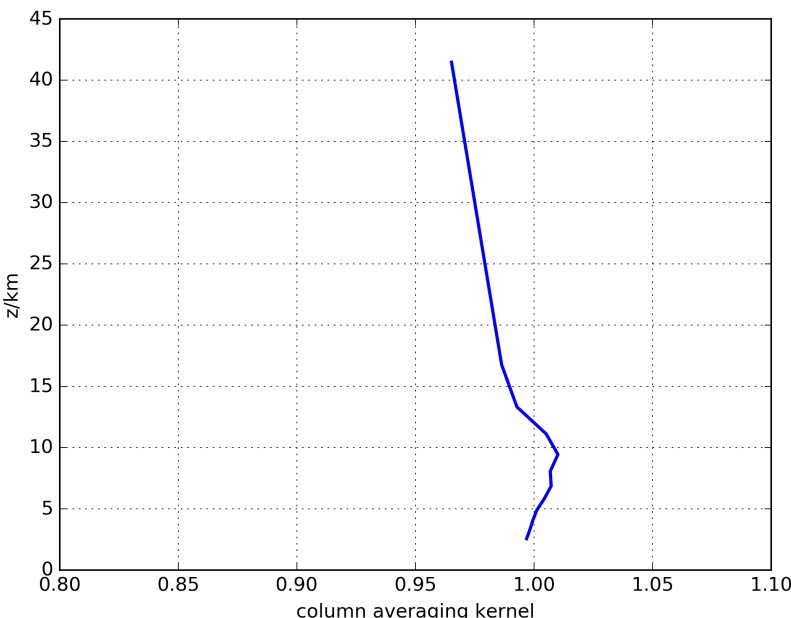

**Fig. 2.** Column averaging kernel for a typical methane retrieval. The retrieval was performed using the simulated NIR and SWIR spectra in Fig. 1.

Here we make use of the fact that, the $H_2O$ column retrieved from the strong band is more sensitive to low altitudes (and thus more sensitive to clouds) than the $H_2O$ column retrieved from the weak band. Based on tests with simulations, we chose 2329–2334 nm as the weak absorption band and 2367–2377 nm as the strong absorption band (see also Scheepmaker et al. (2016)). In Fig. 3, we show the absorption features of $H_2O$ in the SWIR range and highlighted the proposed weak and strong absorption band. We have found that if we filter out data with $|H_2O_{weak} - H_2O_{strong}|/H_2O_{strong} > 0.08$, we filter out most cloudy pixels, see Sect. 4.

Further, before performing any $CH_4$ retrievals, we filter out cases with solar zenith angle (SZA) larger than $70°$, and viewing zenith angle (VZA) larger than $50°$. These thresholds have been derived from simulations and shall be finetuned after launch using real observations.

Butz et al. (2012) identified an a posteriori filter based on retrieved scattering parameters: SWIR aerosol optical thickness $\tau_{swir}$ at 2350 nm, size parameter $\alpha$ and height parameter $z_{aer}$. Following this work, we filter out retrievals with

$$\frac{\tau_{swir} \cdot z_{aer}}{\alpha} > 120 \text{ m}.$$







**Fig. 3.** Simulated reflectance spectrum showing the absorption features of $H_2O$ in the SWIR spectral range for the same scenario as in Fig. 1. The region between the red lines indicate the weak absorption band, while the blue lines enclose the strong absorption band. The difference between the water column retrieved from the weak and strong band is used for the backup cloud filter.

This indicates that the retrieval algorithm has difficulties with scenes that have optically thick scattering layers of large particles at high altitude. Also, we filter out cases with retrieved albedo in the SWIR band smaller than 0.02. More a posteriori filters (e.g. based on goodness of fit) will be determined after launch using real observations.

## 3 Sensitivity studies

Here, we evaluate the sensivity of the retrieved $XCH_4$ forward model error to instrument errors and auxiliary input to the retrieval algorithm (e.g. meteo data). To put the errors in perspective, the S5-P product accuracy of $XCH_4$ was envisioned to be within 2% and the product precision to be within 0.6% (Veefkind et al., 2012). Accuracy is defined as the mean deviation from the truth and precision is the variation due to random processes such as instrument noise. More recently, the requirement





has been slightly reformulated as 1% bias and 1% precision (Hasekamp et al., 2016) to better oppose them with the algorithm performance. From the 1% bias, 0.8% is reserved for forward model errors and 0.6% for instrument related errors.

### 3.1 Synthetic measurements: the global ensemble

We performed detailed sensitivity studies of the $CH_4$ algorithm on a global ensemble of simulated spectra consisting of land-only, clear sky scenes. This ensemble is to a large extent identical to the one used by Butz et al. (2012). It contains realistic aerosol and cirrus loaded scenes for four days, one per season. The treatment of aerosols and cirrus in the simulations are far more complex than in the retrieval forward model, where only one effective aerosol type is considered, see Sect. 2.1.1.

For the simulations, the aerosol physical properties and vertical distributions are derived from the global aerosol model ECHAM5-HAM (Stier et al., 2005) for five different chemical species and on a superposition of 7 log-normal size distributions. The aerosol optical thickness is derived from MODIS observations (Remer et al., 2005). Furthermore, the simulations contains cirrus with optical thickness and vertical distribution based on CALIOP measurements (Winker et al., 2007). Finally,

the surface albedo in the NIR is taken from the MODIS land albedo product in the 841-876 nm channel. For the albedo in the SWIR, the SCIAMACHY surface albedo product at 2350 nm is used (Schrijver et al., 2009). The measurements are simulated for the nadir viewing direction and a solar zenith angle that is representative for TROPOMI with an overpass time of 13:30 local time. We have in total 8633 simulated measurements.

While Butz et al. (2012) investigated only the scattering induced error, we attempt to estimate the total forward model error. Therefore, we increased the inconsistency between the simulation forward model and the retrieval forward model model. The simulations were computed using a line-by-line radiative transfer model, whereas the retrieval method uses the linear k-method. Also, the simulations have a higher vertical and spectral resolution than used in the retrieval. Furthermore, we

have added chlorophyll fluoresence emission. In the simulatons, fluorescence is modelled to have a double Gaussion spectral shape (Guanter et al., 2010), which is different from the linear spectral shape assumed in the retrieval forward model (see Eq. 3). As in the retrieval scheme, we neglect scattering of fluorescence emission for simplicity. The fluoresence at the TOA then becomes

$$F_s(\lambda) = F_{s,755}\Big( \sum_{i=1,2} A_i e^{\frac{-(\lambda-\lambda_i)^2}{\sigma_i^2}} \Big) e^{-\tau_{O_2}(\lambda)/\mu}. \tag{7}$$

For the parameters $A_1$, $A_2$, $\lambda_1$, $\lambda_2$, $\sigma_1$ and $\sigma_2$, we use the same values as in Frankenberg et al. (2012). We only included fluorescence emission for scenes in the global ensemble with $albedo_{NIR}/albedo_{SWIR} > 5$ as a rough selection criterion for regions with vegetation.

After convolving the simulated TOA radiance with the ISRF, the spectra are superimposed with instrument noise from the TROPOMI noise model (Tol et al., 2011). For the NIR, the noise consists

solely of shot-noise, while for the SWIR, the noise is composed of both shot-noise and a signal-





independent term. The corresponding continuum signal-to-noise ratios are 500 in the NIR and 100
in the SWIR for a reference scene with surface albedo $A_s = 0.05$, viewing zenith angle VZA $= 0°$
and solar zenith angle SZA $= 70°$.

The baseline performance of the operational $CH_4$ algorithm is tested on the simulated global
ensemble described above. In Fig. 4, we show a worldmap of the bias $\Delta XCH_{4,\mathbf{F}}$ after applying the
a posteriori filters based on retrieved scattering parameters, albedo and SZA. Note that the error due
to measurement noise $\Delta XCH_{4,\mathbf{y}}$ have been substracted from the total $XCH_4$ error. This is a random
error and is evaluated separately in Sect. 3.3.1 In Fig. 5, the cumulative probablity distribution of the
absolute $XCH_4$ retrieval error is shown (blue line). We get a convergence rate of 99%, and 53% of
the converged retrievals pass the filters. Finally, 94% of the valid retrievals have an absolute error
$< 1\%$. In Fig. 5, we also plotted the cumulative probablity distribution in case fluorescence emission
is not fitted (red line). Then, we have a convergence rate of 95% and 92% of the valid retrievals have
an error $< 1\%$. Thus our retrieval results are improved by fitting fluorescence, which is why we have
included this in the baseline.

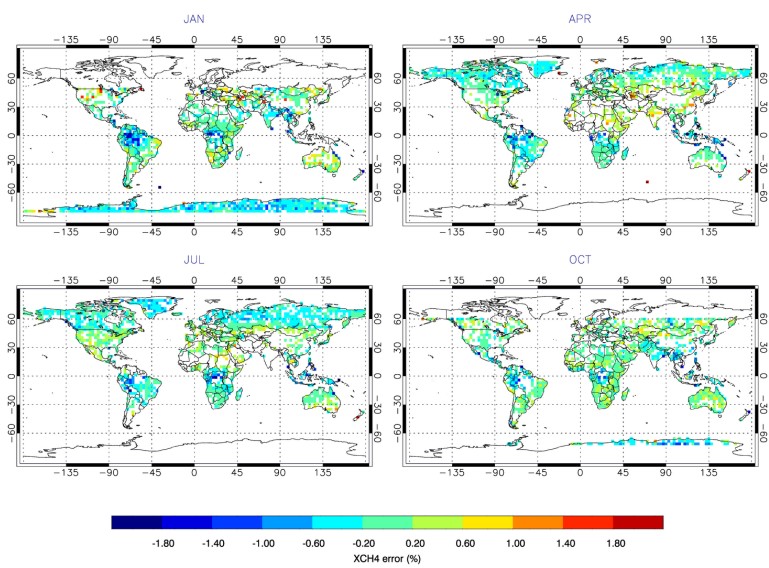

**Fig. 4.** Residual forward model $XCH_4$ errors for the baseline retrieval method for the clear sky global
TROPOMI measurement ensemble after a posteriori filtering.



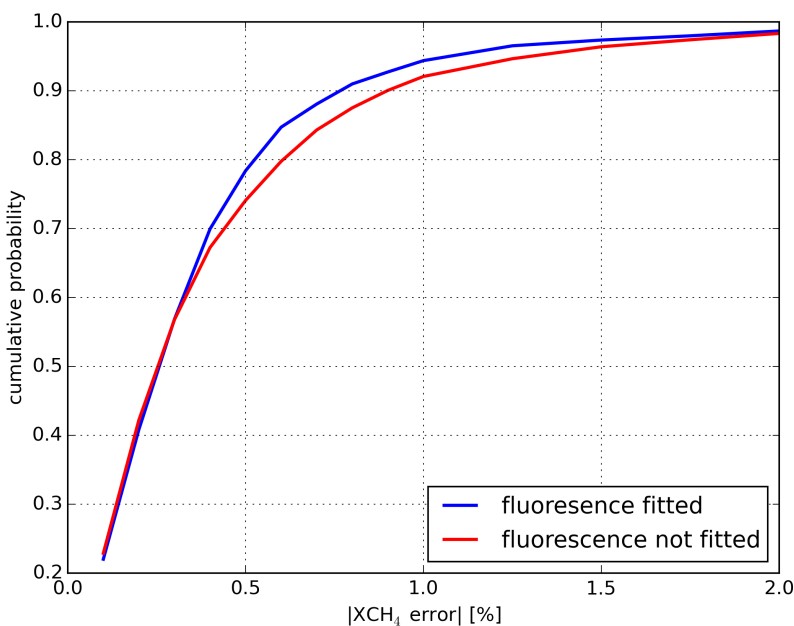

**Fig. 5.** Cumulative probablity distribution of the absolute $XCH_4$ forward model error for the baseline retrieval (blue line) and for the case fluorescence emission is not fitted (red line).

### 3.2 Sensitivity to atmospheric input data

We investigated the effect of imprecise atmospheric input from TM5 and ECMWF on the $CH_4$ retrievals. The results are summarised in Fig. 6.

#### 3.2.1 A priori $CH_4$ profile

For the baseline performance test in Sect. 3.1 we used the true profile of methane $\mathbf{x}_{\mathrm{true}}$ as a priori for the retrievals. The bias $\Delta XCH_{4,\mathbf{F}}$ as defined in Eq. (5) should not depend on the choice of the a priori profile because this effect is accounted for by the averaging kernel. To test this, we take as a priori profile the latitudinal mean with a column deviation up to $\pm 2\%$. To illustrate the effect on the global ensemble, we evaluate the root mean square (RMS) of the $XCH_4$ bias (i.e. the total $XCH_4$ error minus the contribution due to noise) of all retrievals that pass the a posteriori filters. Fig. 6 (first panel, blue line) shows that the a priori $CH_4$ does not influence the retrieval accuracy, in terms of the RMS of the $XCH_4$ bias, nor the stability, in terms of the convergence rate or amount of valid retrievals.



### 3.2.2 A priori H$_2$O profile

To investigate the sensitivity to errors on the assumed H$_2$O profile we follow the same procedure
as in Sect. 3.2.1. The error on the prior H$_2$O profile is established in the same way as for CH$_4$, i.e.
by taking a normalised mean profile per latitude. Note that for H$_2$O, there is an additional (minor)
influence on the retrieval of XCH$_4$ through the dry air column. The H$_2$O column error is varied up
to ± 10%. Fig. 6 (left panel, red line) shows that the prior H$_2$O profile has negligible influence on
the RMS of the XCH$_4$ bias, increasing it with $< 0.01\%$. There is a small effect on the convergence
rate reducing it from 99% to 97% and leading to a reduction of valid retrievals from 53% to 50%.
We note that taking a latitudinal mean profile for H$_2$O represents a worst case in terms of accuracy
for the specific humidity of ECMWF.

### 3.2.3 Pressure

An erroneous pressure affects the retrieval of XCH$_4$ in two ways: first of all, through the pressure
dependence of the cross-sections and, secondly, through the retrieved air column that is used to
convert the CH$_4$ total column to the dry air mixing ratio, XCH$_4$. The latter will introduce a retrieval
error of the same magnitude as the pressure error. To evaluate the net effect of a pressure error, the
prior pressure profile is perturbed with a scaling factor up to $\pm 0.3\%$, corresponding to $|\Delta P_{\mathrm{surf}}| \approx 3$
hPa. We expect a better accuracy from the ECMWF surface pressure together with the Digital
Elevation Map (Salstein et al., 2007; Danielson and Gesch, 2011; Farr et al., 2007). Fig.6 (right
panel, blue line) shows that the increase in the RMS of the XCH4 bias is $< 0.15$ %. There is no
visible effect on the stability of the algoritm.

### 3.2.4 Temperature

An error in the temperature will propagate to the XCH$_4$ retrievals though the temperature dependence
of the cross-sections. To investigate this effect, the temperature profile is offsetted up to $\pm 2$ K. Fig. 6
(right panel, blue line) shows that the increase in the RMS of the XCH4 bias is $< 0.15$ %. There is
a small effect on the stability of the algoritm reducing the convergence rate to 97% and the number
of valid retrievals to 47%.

### 3.3 Sensititvity to instrument errors

We investigated the effect of different possible instrument and calibration errors on the CH$_4$ re-
trievals. The results are summarised in Table 2. Below we discuss each effect separately.

### 3.3.1 Signal to noise ratio

The simulated spectra include instrument noise as described in Sect. 3.1. The precision is given by
the standard deviation of the retrieval noise $\sigma_{\mathrm{XCH}_4}$, see Eq. (6). The worldmap in Fig. 7 shows





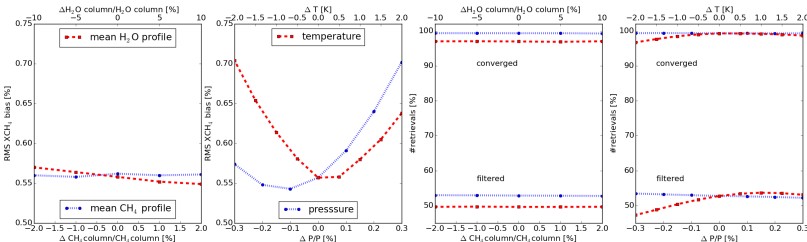

**Fig. 6.** Influence of errors in atmospheric input on the accuracy (panels 1 and 2) and stability (panels 3 and 4) of XCH$_4$ retrievals. The upper and lower x-axis refer to perturbations. Accuracy refers to the XCH$_4$ bias and stability refers to the fraction of converged and valid retrievals. The profiles of methane and water have been perturbed in panels 1 and 3, the pressure and temperature profiles have been perturbed in panels 2 and 4.

the precision relative to the retrieved XCH$_4$. Typically the precision is better than the accuracy. The signal to noise ratio only becomes a limiting factor for scenarios with snow-covered ground and large SZA, which is why we filter for SZA$< 70°$ and albedo$> 0.02$ to keep this error relatively small.

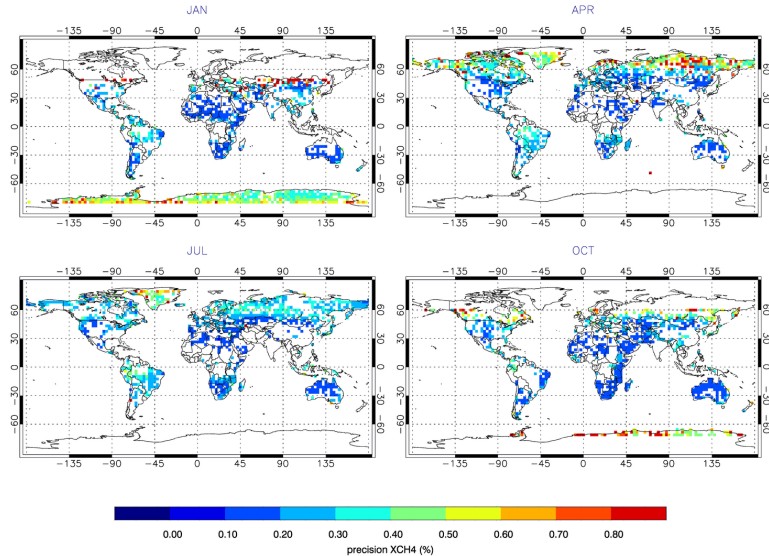

**Fig. 7.** Relative precision of XCH$_4$ due to the instrument noise for the a posteriori filtered dataset for the clear sky ensemble in Fig. 4



### 3.3.2 Instrument spectral response function

The synthetic measurements were created by convolving the underlying line-by-line spectra with

a Gaussian ISRF with a full width half maximum (FWHM) of 0.38 nm in the NIR band and 0.25 nm in the SWIR band. The mission requirements state that the ISRF shall be known within 1% of its maximum (Langen et al., 2011). This is achieved approximately by varying the FWHM by 1%. Table 2 give the results for retrievals with an assumed error in the FWHM. We find that the retrievals are mostly sensitive to the accuracy of the ISRF in the SWIR band, leading to an increase

of the RMS $XCH_4$ bias from 0.56% (baseline) to 0.67%. We note that of all the instrument errors investigated here, the ISRF gives the largest error contribution to the methane retrievals. Thus our study indicates that accurate calibration of the ISRF should have high priority.

### 3.3.3 Spectral calibration

According to the instrument requirements, the centre wavelengths of spectral channels are known

within 2 picometres. For our global ensemble, a spectral shift of 2 pm has negligible effect on the error characteristics. Here, we evaluate the effect of a wavelength shift of 1/20 of the spectral sampling distance, i.e. 10 pm and 5 pm for the NIR and SWIR band, respectively. The reference retrieval fits a spectral shift. To test this fitting option, the synthetic spectra were shifted with a constant wavelength:

$$\lambda_{k,\mathrm{meas}} = \lambda_k + \Delta\lambda \qquad (8)$$

where $\lambda_k$ is the real wavelength, $\lambda_{k,\mathrm{meas}}$ is the measured wavelength at pixel k and $\Delta\lambda$ the spectral shift. In Table 2, results are given for the case that a spectral shift is not fitted and is fitted (between brackets). When fitted, the performance is as good as for the reference retrieval, i.e. simulations without an error in the spectral position. This is as expected and indicates that the spectral shift

fitting is robust.

Optionally a wavelength dependent shift, $\Delta\lambda_{\mathrm{squeeze}}$, can be fitted. To test this fitting option, the synthetic wavelength grid was "squeezed":

$$\lambda_{k,\mathrm{meas}} = \lambda_k + \Delta\lambda_{\mathrm{squeeze}} \times (\lambda_k - \lambda_{\mathrm{mid}})/(\lambda_{\mathrm{end}} - \lambda_{\mathrm{mid}}) \qquad (9)$$

where $\lambda_{\mathrm{end}}$ and $\lambda_{\mathrm{mid}}$ are the wavelengths at the end and middle of the band, repectively. Table 2

shows the performance of retrievals with this assumed error on the measured wavelength grid. Since a spectral squeeze is a second order effect and has a much smaller impact on the $XCH_4$ retrievals than a spectral shift, it is not fitted in the baseline. However, our results show that, if needed, the option to fit a spectral squeeze can be used reliably.



**Table 2.** Effect of instrument calibration errors on convergence rate, fraction of valid retrievals after filtering, and RMS values of $XCH_4$ bias and precision for the global ensemble. Note that all sensitivities include the baseline error. The terms between brackets are for the cases where the relevant quantity is also retrieved. For each instrument calibration error, multiple simulation runs were performed with all combinations of errors in NIR and SWIR channels. The results shown here correspond to the runs with poorest performance in terms of the RMS error.

| | Convergence | valid retrievals | RMS of $XCH_4$ bias | RMS of $XCH_4$ precision |
|---|---|---|---|---|
| baseline | 99% | 53% | 0.56% | 0.43% |
| $\Delta FWHM_{NIR} = -1\%$ | 99% | 53% | 0.56% | 0.43% |
| $\Delta FWHM_{SWIR} = -1\%$ | 99% | 49% | 0.67% | 0.44% |
| $\Delta FWHM_{NIR/SWIR} = -1\%$ | 99% | 49% | 0.67% | 0.44% |
| $\Delta \lambda_{shift,NIR} = -10$ pm | 97% | 48% | 0.57% | 0.42% |
| $\Delta \lambda_{shift,SWIR} = 5$ pm | 99% | 50% | 0.99% | 0.42% |
| $\Delta \lambda_{shift,NIR/SWIR} = -10$ pm/5 pm | 99% (99%) | 46% (53%) | 1.02% (0.56%) | 0.41% (0.43%) |
| $\Delta \lambda_{squeeze,NIR} = 10$ pm | 99% | 53% | 0.56% | 0.42% |
| $\Delta \lambda_{squeeze,SWIR} = 5$ pm | 99% | 52% | 0.62% | 0.42% |
| $\Delta \lambda_{squeeze,NIR/SWIR} = 10$ pm/5 pm | 96% (99%) | 53% (53%) | 0.63% (0.56%) | 0.42% (0.43%) |
| $I_{offset,NIR} = -0.1\%$ | 99% | 53% | 0.57% | 0.43% |
| $I_{offset,SWIR} = 0.1\%$ | 99% | 50% | 0.58% | 0.43% |
| $I_{offset,NIR/SWIR} = -0.1\%/0.1\%$ | 99% | 50% | 0.59% | 0.43% |
| $G_{NIR} = 1.02$ | 99% | 53% | 0.56% | 0.43% |
| $G_{SWIR} = 1.02$ | 99% | 53% | 0.58% | 0.42% |
| $G_{NIR} = G_{SWIR} = 1.02$ | 99% | 52% | 0.58% | 0.42% |

### 3.3.4 Radiometric offset: additive factor

The effect of an unknown systematic offset in the Earth radiance is investigated. The offsets in the NIR and SWIR bands are independently varied with $\pm 0.1\%$ of the continuum. Table 2 shows the effect of a radiometric offset to the $XCH_4$ retrievals. We note that a radiometric offset in the SWIR band causes a larger $XCH_4$ error than an offset in the NIR band. The latter is partly compensated by the retrieved fluorescence.

### 3.3.5 Radiometric gain: multiplicative factor

The absolute radiometric accuracy of the measurement of the Earth spectral radiance shall be better than 2% according to the system requirements. To investigate the effect of such an error, the synthetic spectra were multiplied with a scaling factor $G$. Table 2 shows that there is negligible effect of an error of 2% in radiometric gain. This error is largely compensated by the retrieved surface albedo. It



follows that interaction between surface albedo and aerosols has a negligible impact for gain errors
$< 2\%$.

### 3.4 Heterogenous slit illumination

For the two-dimensional TROPOMI push-broom spectrometer, light in across-slit dimension is dis-
persed by the instrument grating in order to spectrally resolve the received signal. The along-slit di-
rection is aligned across flight direction to achieve the desired spatial resolution. For a homogeneous
illumination of the instrument slit, the spectral instrument response is characterised extensively dur-
ing the pre-flight calibration of the TROPOMI instrument and it is used as baseline to simulate the
TROPOMI radiometric measurement in our retrieval. In space, however, the instrument slit will be
illuminated inhomogeneously due to ground scene heterogeneities on scales smaller than the instru-
ment's field of view. Inhomogeneous illumination across the slit leads to a distortion of the ISRF as
described by Noel et al. (2012), Caron et al. (2014) and Landgraf (2016) and can affect the retrieval
accuracy of the TROPOMI methane product. Small-scale heterogeneities of the ground scene are
generally caused by spatial variations of surface reflection and by broken clouds. Because of our
strict cloud filtering, spatial variations in surface reflection are the only cause of methane retrieval
biases due to inhomogeneous slit illumination. To evaluate the effect of surface scene heterogeneity
on our $XCH_4$ product, we employ the instrument model described by Landgraf (2016) for both the
NIR and SWIR bands of our retrieval. Furthermore, we use a high spatial resolution MODIS albedo
map for the $50 \times 50 \text{ km}^2$ marsh region in central Siberia with structures in the surface reflection due
to ponds, shown in Fig. 8. Depending on the scene heterogeneity in the flight direction, the $XCH_4$
error shows an oscillation structure with a maximum amplitude $\leq 0.4$ %, a standard deviation of
0.12 % and a mean error of -0.01 %. For a particular temporal and spatial sampling of the scene,
a pseudo-random scatter is introduced to the $XCH_4$ product. This means that overall the effect can
be considered small. One may consider this error as a limitation when interpreting very localised
sources in surroundings of heterogenous surface reflection, but for most applications some averaging
either in time or space will be done which reduces this error.

### 4  Cloud filtering

The global ensemble from Butz et al. (2012) as used in Sect. 3 cannot be used to evaluate the cloud
filters, because it consists purely of cloud free scenes. Therefore, the cloud filters are tested using a
new ensemble of synthetic measurements, which also include scenes with water clouds.

### 4.1  Synthetic measurements: the TROPOMI test orbit

To test the performance of the proposed backup cloud filter, we have constructed synthetic TROPOMI
L1B radiance spectra for an entire orbit which passes over Africa. We used realistic viewing geome-





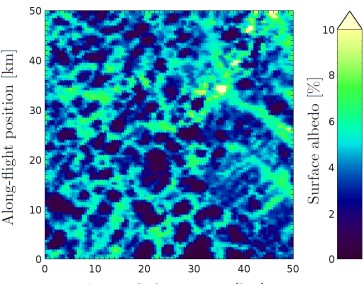 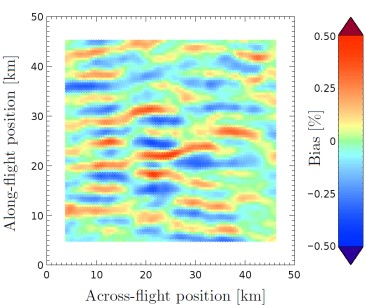

**Fig. 8.** Methane bias due to heterogenous slit illumination for spatially varying surface reflection over a marsh scene at Siberia close to the river Ob at latitude 62.8°N and longitude 72.1°E. Measurement simulations are performed with the instrument model by Landgraf (2016) for an instantaneous field of view of 3.4 km across the slit and 7.0 km along the slit.

tries from a TROPOMI orbit simulator provided by the Royal Netherlands Meteorological Institute (KNMI). The meteorological data is from ECMWF. We used CO and $CH_4$ model profiles from TM5
(Houweling et al., 2014). The surface elevation comes from a digital elevation map constructed by KNMI based on USGS (Danielson and Gesch, 2011) and NASA data (Farr et al., 2007). The aerosol and cirrus properties and surface albedos are taken from the global ensemble, which means that these are the same for all TROPOMI groundpixels within a $\sim 3° \times 3°$ latitude×longitude box. While the global ensemble used for the sensitivity studies is land-only and clear sky, the test orbit
contains cloudy scenes with cloud information (cloud fraction, cloud optical thickness and cloud top height) from MODIS, over land and ocean. For reference, the cloud fraction used in the simulations are shown in Fig. 9, upper left. We note that 28% of the TROPOMI groundpixels are fully cloud free for this test orbit. Globally, one would expect on average 20% cloud free pixels (Krijger et al., 2007). Resampling of the auxiliary data on the TROPOMI groundpixels have been performed using
the Multi-Instrument Preprocessor (MIPrep) developed at SRON.

### 4.2 Performance of cloud filters

First, we show the performance of the $XCH_4$ retrievals on the test orbit when no a posteriori data filtering is applied, only a priori filtering of ocean pixels and SZA> 70° and VZA> 50°, see Fig. 9, upper right. It is clear that cloud contaminated measurements lead to large $XCH_4$ errors ($> 2\%$).
Assuming that we have cloud data from VIIRS, we would then be able to filter out the cloudy pixels almost perfectly. To illustrate the effect, we filtered out pixels with cloud fraction $> 0.02$ in Fig. 9, lower left. Note that in this case we have also applied a posteriori filtering based on retrieved scattering parameters and albedo. One is then left with valid retrievals of $\sim 3\%$ of all simulations in the test orbit.
In comparison, the performance of the backup cloud filter based on the difference between the





**Table 3.** Error statistics of XCH$_4$ retrievals from the L1B orbit using different cloud filters in comparison to the global clear sky orbit. The performance of the MODIS filter is expected to be comparable with the operational cloud filter using VIIRS data.

| | retrievals with $\|XCH_4\ error\| < 1\%$ | retrievals with $\|XCH_4\ error\| < 0.5\%$ | RMS of XCH$_4$ bias | RMS of XCH$_4$ precision |
|---|---|---|---|---|
| global clear sky ensemble | 94% | 78% | 0.56% | 0.43% |
| L1B orbit with MODIS filter | 96% | 80% | 0.56% | 0.31% |
| L1B orbit with backup cloud filter | 94% | 79% | 0.71% | 0.27% |

H$_2$O column retrieved from strong and weak bands (see Section 2.3) is shown in Fig. 9, lower right. The backup cloud filter removes most cloudy pixels, but some remain. In Table 3 and Fig.10, the statistics of XCH$_4$ retrievals on the orbit are summarised. After cloud filtering with MODIS data (representative for operational VIIRS data), the results for the test orbit are comparable to the clear-sky global ensemble. However, the backup cloud filter is less effective. The RMS of the XCH$_4$ bias is then 0.71% instead of the 0.56% that is expected for the operational VIIRS cloudmask.

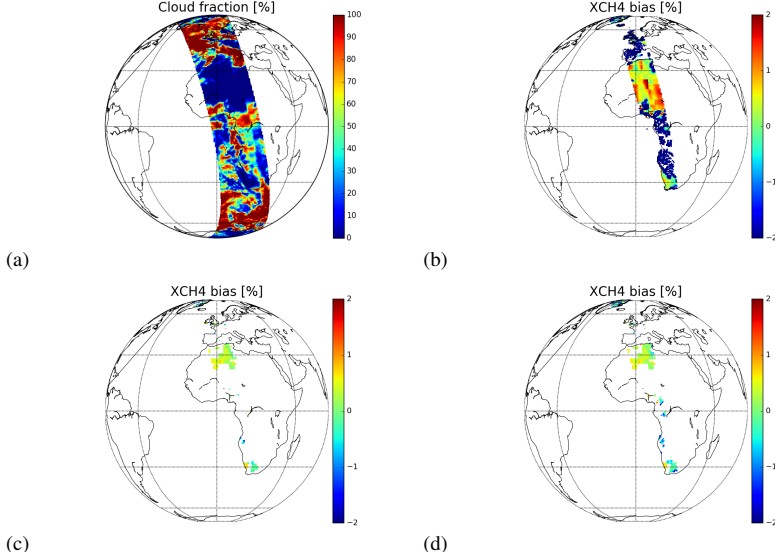

**Fig. 9.** Simulations of a TROPOMI orbit. Panel (a) shows MODIS cloud fraction resampled on the orbit's groundpixels. Panel (b) gives the XCH$_4$ bias of all processed pixels that converged. Panel (c) and (d) give the valid retrievals after cloud filtering with MODIS data and the backup cloud filter, respectively. Here, we also applied the a posteriori filter based on retrieved scattering parameters and albedo.



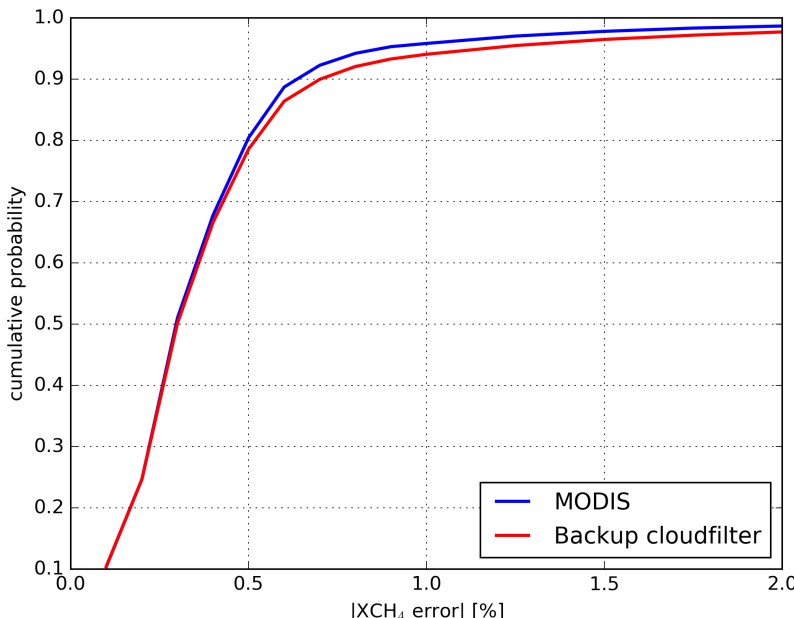

**Fig. 10.** Cumulative probablity distribution of the absolute XCH$_4$ bias for the simulated level 1b orbit. Cloud-contaminated measurements are filtered using MODIS data (blue line) or the backup cloud filter (red line).

## 5  Conclusions

This paper describes the algorithm baseline of the operational methane retrievals from the S5-P measurements. The level 2 product includes the column-averaged dry air mixing ratio XCH$_4$, the column averaging kernel and the noise standard deviation. In order to account for the effect of aerosols and cirrus, the developed algorithm retrieves the methane column simultaneously with effective scattering parameters related to particle amount, size and height distribution. The choice of scattering parameters reflects the information content of the measurements as close as possible. The retrieval algorithm uses the radiance and irradiance measurements in the SWIR 2305–2385 nm band and additionally in the NIR band between 757–774 nm (O$_2$-A band). The forward model of the retrieval algorithm uses online radiative transfer calculations, fully including multiple scattering in an efficient manner. Absorption cross sections of the relevant atmospheric trace gases and optical properties of aerosols are calculated from lookup tables. The inversion is performed using Phillips-Tikhonov regularisation in combination with a reduced step size Gauss-Newton iteration scheme.



To test the developed algorithm we generated two ensembles of simulated measurements that cover the range of scenes that will be likely encountered by the S5-P instrument; one clear sky global ensemble and one test orbit containing cloud-contaminated measurements. Overall, the developed algorithm performs well in correcting for the effect of aerosols and cirrus clouds on the retrieved

$XCH_4$. For both ensembles, $\sim 80\%$ of the cases have an $XCH_4$ error $< 0.5\%$ and $\sim 95\%$ have an error $< 1\%$. To achieve this a priori filtering of cloud contaminated scenes and a posterirori filtering based on retrieved parameters are necessary. We illustrated the performance of the proposed backup cloud filter based on retrievals of $H_2O$ from weak and strong absorption bands in the SWIR under the assumption of a non-scattering atmosphere. It should be noted that the cloud filter based on S5-P

measurements itself is less efficient than the VIIRS cloudmask for water clouds.

Apart from forward model errors induced by aerosols, we also studied effects of model errors in temperature, pressure, and water vapour profiles. We expect to stay within product requirements for errors in input profiles of water, pressure and temperature below 10%, 0.3% and 2 K, respectively. Another relevant source of errors to the $CH_4$ data product could be spectroscopic errors. This has

been studied in detail by Galli et al. (2012) and Checa-Garcia et al. (2015). Note that a study is ongoing to improve the spectroscopic data for TROPOMI's SWIR spectral range (Loos et al., 2015). Concerning instrument errors, we found that the most critical error source is an error in the ISRF in the SWIR band. To conclude, we have shown that for a compliant instrument our algorithm provides a methane product that meets the requirements.

*Acknowledgements.* We thanks M. Sneep and S. Houweling for providing us with auxiliary data needed to simulate TROPOMI measurements. This research has been (in part) funded by the TROPOMI national program from the Netherlands Space Office (NSO). A.B. is supported by Deutsche Forschungsgemeinschaft through the Emmy-Noether programme, grant BU2599/1-1 (RemoteC).



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
