# Peer review of "The operational methane retrieval algorithm for TROPOMI"

_Atmospheric Measurement Techniques, 2016_

## Referee Comment (RC1) · Anonymous Referee #1 · 25 May 2016

Reviewer's comments on: "The operational methane retrieval algorithm for TROPOMI" by H. Hu, et al.

General Comments

This manuscript by Hu et al. describes the retrieval algorithm designed to determine methane column-averaged dry air volume mixing ratio from TROPOMI spectra measured in two bands. Much of the first half of the paper (Sections 1 and 2) is background material and was presented previously in the paper by Butz et al. (2012). The new simulations and sensitivity studies presented in Sections 3 and 4 address the expected performance of the algorithm in 'real world' situations. A variety of degrading effects are considered, including aerosols, clouds, forward model error, instrument SNR, errors in assumed temperature and water vapor profiles, and calibration. Results of individual experiments are mainly evaluated in terms of (1) the overall fraction of 'valid' retrievals not rejected by filtering and (2) root-mean-square bias and precision statistics.

Overall, this paper will be quite valuable to future users of TROPOMI products wanting to understand how to properly interpret these products. For the most part, the analysis methods used are appropriate, and the results are properly interpreted. However, as described below, I believe the paper ignores at least one significant source of potential retrieval error, and also makes no attempt to analyze the scenarios which lead to rejected observations (i.e., in the filtering stage). In the revised manuscript, the authors should specifically address these issues, and discuss how these issues will affect potential users.

Surface albedo is strongly wavelength-dependent, for most (if not all) land surface types, however, the TROPOMI algorithm assumes a simple first-order spectral dependence across the NIR and SWIR bands. For example, green vegetation exhibits a sharp increase in reflectance near the TROPOMI NIR band (757-774 nm) and the reflectance of vegetation is likely not linear across this spectral band. Various types of minerals and soils also exhibit complex spectral dependence of surface reflectance in both the NIR and SWIR bands. Why was this effect not represented in the simulations?

As described in Section 2.3, filters are used to discard those observations where the reliability of the methane retrieval is expected to be poor. Globally, approximately one half of the observations will be rejected. This will include scenes where aerosols exhibit high optical depth, large particles, and are located at high altitude. Thus, it should be expected that rejected observations will not be random, but will probably occur much more frequently in certain geographical regions (and during certain seasons) compared to others. This is extremely important information for potential users and should be specifically addressed in the revised manuscript. Where will the TROPOMI methane retrieval algorithm generally be useful and where won't it be useful? For example, it appears that there are very few successful retrievals over China and India (in Fig. 7), but it is not clear if this might just be due to the lack of ensemble 'test profiles' located there. Maps of the locations of rejected retrievals should be shown.

Minor Revisions and Technical Corrections

p. 5, line 125. Does the entire forward model grid really change as p_sfc changes (in order to create 36 layers with equal pressure widths), or is there a fixed pressure grid above the surface?

p. 5, l. 139 (and following lines). Various assumptions are made in the aerosol model, including values for r_1, r_2, and the aerosol complex refractive index. How is it known that these are reasonable values?

p. 6, l. 157. Define 's' in Eq. 3.

p. 6, l. 161. What is the relationship between the 12-layer retrieval grid and the 36-layer forward model grid?

p. 6, l. 162. It appears that the retrieval algorithm simultaneously retrieves both methane and CO total column amounts. Is this CO total column value just a diagnostic for the methane retrieval, or is this the official TROPOMI CO product?

p. 8, l. 200. Suggest rewording phrase 'have sensitivity down to the ground'

p. 8, l. 211. Is there a reference for this equation for total column retrieval uncertainty? Please show how this equation was derived. As shown in Rodgers' book (Section 4.3, 'Best Estimate of a Function of the State Vector'), the measurement error for total column will involve the matrix Sx as well as the linear operator which relates the retrieved profile and the total column.

p. 8 l. 213. It should be noted somewhere that the simulations will not represent all types of forward model errors, such as errors in the underlying spectroscopic database.

p. 11, l. 251. Meaning of the phrase 'to better oppose them with the algorithm performance' is unclear to me.

p. 11, l. 256. Properties of the ensemble are very important to interpreting the simulation experiments. How are the ensemble profiles distributed geographically? What is the range of aerosol optical depths in the ensemble?

p. 11, l. 265. Clarify that simulations assume constant surface albedo within NIR and SWIR bands.

p. 12, l. 286. Are these SNR values really 'design goals' or are they based on actual instrument engineering data?

p. 13, Sec. 3.2.1, first paragraph. Is the a priori methane profile experiment really done using the 'latitudinal mean' (based on an average over all latitudes) or 'zonal mean' (based on an average over longitude)? In either case, please provide some more detail, such as the width of the latitude or longitude bins used to calculate the mean.

p. 21, Conclusion. The conclusion should include information about the filtering algorithm, and what scenarios (geographical regions and seasons) most often lead to filtered retrievals.

---

## Referee Comment (RC2) · Anonymous Referee #2 · 23 Aug 2016

General Comments This paper introduced the operational XCH4 retrieval algorithm in high precision and accuracy, which is valuable for application of CH4 monitoring using the Sentienl-5 precursor satellite, and its performance tested on realistic simulated measurements. There are sensitive studies on XCH4 retrieval errors caused by atmospheric scattering properties, atmospheric data and instrument calibrations. It is useful in future improvement of the retrieval algorithm. This paper is valuable to be published, but need some modification. The author should pay more attention on the following point,

1. As the retrieval algorithm is firstly introduced in this paper, the authors need to explain the detailed description of the retrieval algorithm. For example, in Table 1, the authors should summarize the detailed a priori information, its structure and units as well as the state vector elements used in the algorithm.

1. As the retrieval algorithm is firstly introduced in this paper, the authors need to

explain the detailed description of the retrieval algorithm. For example, in Table 1, the authors should summarize the detailed a priori information, its structure and units as well as the state vector elements used in the algorithm.

2. The authors should highlight the advantage of the retrieval algorithm, as compared with other algorithms for CH4 monitoring.

3. As described in Section 2.1.1, the aerosol type in this retrieval algorithm is characterized by the refractive index and size distribution, which are assumed and fixed parameters in the algorithm. However, aerosols information is the one of the most important factors on XCH4 retrievals and thus the authors should consider the effects of assumed aerosol information on XCH4 retrievals.

4. There is unclear description on the following point, A. In Section 2.3, authors describe the filtering criteria to remove the retrievals with bad quality. B. In line 233, authors applied the filtering criteria for the ratio of retrieved H2O column between at weak and strong band. Authors should describe the reason why the ratio is lower than 0.08.

5. In Section 3, the authors described the sensitivity of XCH4 retrievals to atmospheric input data and instrument errors. The authors would better describe about the sensitivity to assumed information in the algorithm. In addition, the authors would better show the sensitivity to aerosol information, such as aerosol column, aerosol size parameter and aerosol height parameter.

6. In Section 4, cloud information derived from the MODIS products, such as cloud fraction, cloud optical thickness and cloud top height. The authors should briefly explain the method to obtain the cloud information.

7. In Section 2.3, the authors described the method to filter out scenes including cloud screening. Moreover, in Section 4, the authors also described about cloud screening method and its performance in terms of comparison of two filtering method. The methods are mixed in these sections and is unclear. Please make sure the filtering criteria for post-screening and cloud screening in individual section if you want to simultaneously show its performance.

Minor Comments

1. In Fig. 3, please add a legend label.

2. In Fig. 4, is it monthly mean value of retrievals? Please show the period that you have these results. Also, you can plot again with large legend.

3. Table 2 shows the summarized the effect of instrument calibration errors on the XCH4 bias and precision. The authors would better summarize the effect of atmospheric input data and assumed scattering simultaneously.

4. In Fig. 9, authors describe the panel (c) and (d) give the valid retrievals after cloud filtering with MODIS and the backup cloud filters. The authors should revised that panel give the XCH4 biases of the valid retrievals after filtering.

5. In 471, please check spelling in the sentence

---

## Author Comment (AC1) · 19 Sep 2016

**Author's reply to Anonymous Referee #1**

We would like to thank referee #1 for his/her comments that helped improve the quality of the paper. Below we addressed the comments one-by-one; comments from the referee are typeset in italic, our replies are in normal font, and our changes in manuscript are in blue. Line, page and figure numbers in the referee's comments refer to the original manuscript, whereas in our reply we give page and line numbers that refer to the revised manuscript.

**General Comments**

*1) Surface albedo is strongly wavelength-dependent, for most (if not all) land surface types, however, the TROPOMI algorithm assumes a simple first-order spectral dependence across the NIR and SWIR bands. For example, green vegetation exhibits a sharp increase in reflectance near the TROPOMI NIR band (757-774 nm) and the reflectance of vegetation is likely not linear across this spectral band. Various types of minerals and soils also exhibit complex spectral dependence of surface reflectance in both the NIR and SWIR bands. Why was this effect not represented in the simulations?*
Regarding the TROPOMI CH4 retrieval algorithm, the baseline is to retrieve the surface albedo up to first-order spectral dependence, however the option exist to retrieve higher-order spectral dependence. This option has been tested with GOSAT observations and it will be examined also for TROPOMI when real data is availiable. To clarify this, we have added the following sentences on page 11, line 292:
For simplicity, we assume a constant surface albedo within the NIR and SWIR band in our simulations. We expect no difficulties to fit more realistic spectral dependent surface albedo based on our experience with real GOSAT data.
and on page 6, footnote:
Note that the algorithm has the option to fit higher order spectral dependence of the surface albedo. It will be investigated on real data whether this is needed.
Further, the surface albedo used in the simulations are taken from MODIS and SCIAMACHY albedo product for the NIR and SWIR spectral bands, respectively. As far as we are aware of there exist no global database of the spectral dependence of the albedo in the NIR and SWIR window, hence the simulations assume a constant albedo within a band.

*2) As described in Section 2.3, filters are used to discard those observations where the reliability of the methane retrieval is expected to be poor. Globally, approximately one half of the observations will be rejected. This will include scenes where aerosols exhibit high optical depth, large particles, and are located at high altitude. Thus, it should be expected that rejected observations will not be random, but will probably occur much more frequently in certain geographical regions (and during certain seasons) compared to others. This is extremely important information for potential users and should be specifically addressed in the revised manuscript. Where will the TROPOMI methane retrieval algorithm generally be useful and where won't it be useful? For example, it appears that there are very few successful retrievals over China and India (in Fig. 7), but it is not clear if this might just be due to the lack of ensemble 'test profiles' located there. Maps of the locations of rejected retrievals should be shown*
Although, it is in general true that geographical regions with high aerosol loads have more chance to be filtered out, we prefer not to draw too strong conclusions on which regions and to what extent based on our global ensemble because it was not designed for this purpose. Rather, it was designed to test the mean retrieval performance per season based on a representative day. So, detailed information on regional scales down to the individual TROPOMI pixel size are not given by the

ensemble. First of all, the spatial resolution used in our study (~3x3 degrees) is much coarser than TROPOMI's resolution, thus where in our study whole regions might be filtered out, TROPOMI may still have useful observations. Secondly, since our simulations only represent one day per season with a mean atmospheric state, and one should therefore be careful in generalizing to observations entire seasons. To summarize, it is not that not whole regions will be filtered out but only that certain regions will have less days with success retrievals during certain seasons.

With this in mind and to illustrate which regions are more likely to be filtered out by our method, we have added the worldmap of unfiltered retrievals in Fig. 4a on page 14. and added a paragraph on page 13, line 332:
Note further that most of the rejected retrievals are found in the desert and dust regions in Northern Africa and Central Asia in April and July where our simulations capture dust storm events. For real TROPOMI measurements though, we will less likely filter out large regions because TROPOMI has higher spatial resolution than our simulations. Also, our simulations only represent one day per season with a mean atmospheric state. Therefore our results merely indicate that certain regions will have less days with successful retrievals in certain seasons, not that that entire regions will be rejected.

Furthermore, on closer investigation, we found that Fig. 7 was generated with wrong (stricter) filtering that lead to fewer points than shown in Fig. 4. We fixed this and replaced Fig. 7 which is now consistent with Fig, 4 in terms of plotted points. The situation is thus less severe as we have more successful retrievals in China and India than shown before.

**Minor Revisions and Technical Corrections**

*p. 5, line 125. Does the entire forward model grid really change as p_sfc changes (in order to create 36 layers with equal pressure widths), or is there a fixed pressure grid above the surface?*
It is as explained on page 5, line 126, so no changes are made in the manuscript. To answer the reviewer's question: the forward model pressure grid will differ per retrieval in case the surface pressure is different. During a single retrieval though, the pressure grid remains fixed as the algorithm does not retrieve the surface pressure in the baseline configuration.

*p. 5, l. 139 (and following lines). Various assumptions are made in the aerosol model, including values for r_1, r_2, and the aerosol complex refractive index. How is it known that these are reasonable values?*
The aerosol complex refractive index we use is an average complex refractive index weighted by the relative mass contribution of the individual aerosol chemical types in the ECHAM5-HAM model per spectal window. It has been shown by Butz et al 2010 that the results are not sensitive to the exact choice of the refractive indices.
To clarify the above, we have added on page 5, line 143:
which are averaged values derived from the ECHAM5-HAM model  (Stier et al. 2005).
and on page 6, line 154:
Butz et al. 2010 found that the the exact choice of the fixed-value parameters such as the refractive indices or the width of the height distribution are not affecting the retrievals significantly.
Concerning the values for r_1 and r_2, these are from Mischenko et al 1999 as referenced in the paper on page 5, line 144.

*p. 6, l. 157. Define 's' in Eq. 3.*
We have added a sentence on page 6, line 164:

$F^{surf}_{s,755}$ and s represent the chlorophyll emission at 755 nm and its spectral slope, respectively.

*p. 6, l. 161. What is the relationship between the 12-layer retrieval grid and the 36-layer forward model grid?*
They share the same interfaces, thus there are three sublayers of the model atmoshere grid in the retrieval grid. This is now described explicitly in Section 2.1.2, page 6, line 167
The 12 retrieval layers are related to 36-layer model atmosphere by the shared interfaces, i.e. each retrieval layer is divided in three sublayers for the forward model calculations.

*p. 6, l. 162. It appears that the retrieval algorithm simultaneously retrieves both methane and CO total column amounts. Is this CO total column value just a diagnostic for the methane retrieval, or is this the official TROPOMI CO product?*
Our retrieved CO column is not the official TROPOMI CO product, but indeed a diagnostic of the CH4 retrieval. The CO retrieval algorithm for TROPOMI is described in a separate publication by Landgraf et al. in this special issue. We have added a sentence in the paper to clarify this, page 7, line 178
Further, we like to mention that our retrieved CO column should be regarded as a diagnostic for the CH4 retrieval. The official TROPOMI CO product has a dedicated retrieval algorithm described by Landgraf et al 2016.

*p. 8, l. 200. Suggest rewording phrase 'have sensitivity down to the ground'*
See page 8, line 215, rephrased to:
This illustrates that the retrieval of methane columns from the SWIR has a nearly ideal sensitivity to methane in the troposphere and the tropospheric boundary layer.

*p. 8, l. 211. Is there a reference for this equation for total column retrieval uncertainty? Please show how this equation was derived. As shown in Rodgers' book (Section 4.3, 'Best Estimate of a Function of the State Vector'), the measurement error for total column will involve the matrix Sx as well as the linear operator which relates the retrieved profile and the total column.*
We have added a reference to Rodgers (2000) on page 9, line 225. Note that Eq, (7) follows directly from Rodger's equation for the error covariance (h^T S h) in Section 4.3 after equation (4.48), where the linear operator that relates XCH4 to the state vector is given in our paper in Eq. (4).

*p. 8 l. 213. It should be noted somewhere that the simulations will not represent all types of forward model errors, such as errors in the underlying spectroscopic database.*
This was already mentioned in in Section 5, but we also changed and added sentences on page 12, line 299:
While Butz et al. (2012) investigated only the scattering induced error, we increased the inconsistency between the simulation forward model and the retrieval forward model model. We have attempted to include the most important contributions to the forward model error, except for errors due to the underlying spectroscopic database which have been investigated elsewhere, see Galli et al. (2012) and Checa-Garcia et al. (2015).

*p. 11, l. 251. Meaning of the phrase 'to better oppose them with the algorithm performance' is unclear to me.*
We removed this phrase to avoid confusion.

*p. 11, l. 256. Properties of the ensemble are very important to interpreting the simulation experiments. How are the ensemble profiles distributed geographically? What is*

*the range of aerosol optical depths in the ensemble?*

As mentioned at the beginning of Section 3.1, the ensemble is already described in detail by Butz et al 2012, we therefore found it unnecessary to repeat exaclty the same plots. However, we included a reference to the relevant figures at the appropriate places, page 12, line 295:

We refer to Fig.2 and Fig. 3 of Butz et al. 2012 for the geographical distribution of the total optical thickness and surface albedo, respectively, used in the simulations.

*p. 11, l. 265. Clarify that simulations assume constant surface albedo within NIR and*
*SWIR bands.*

We have added the sentence on page 12, line 292:

For simplicity, we assume a constant surface albedo within the NIR and SWIR band in our simulations. We expect no difficulties to fit more realistic spectral dependent surface albedo based on our experience with real GOSAT data.

*p. 12, l. 286. Are these SNR values really 'design goals' or are they based on actual*
*instrument engineering data?*

They are requirements/design goals, as mentioned on page 11, line 276.

*p. 13, Sec. 3.2.1, first paragraph. Is the a priori methane profile experiment really*
*done using the 'latitudinal mean' (based on an average over all latitudes) or 'zonal*
*mean' (based on an average over longitude)? In either case, please provide some*
*more detail, such as the width of the latitude or longitude bins used to calculate the*
*mean.*

Indeed, we should have use 'zonal mean'. This has been corrected and clarified, e.g. on page 13, line 346:

zonal mean, where we averaged over all longitudes within a 2.79 degree latitude bin,

*p. 21, Conclusion. The conclusion should include information about the filtering algorithm,*
*and what scenarios (geographical regions and seasons) most often lead to*
*filtered retrievals.*

W have added an paragraph on page 13, see our reply to General Comment 2. where we also explain why we are careful not to draw too strong conclusions on the filtering with respect to geographical regions and seasons. Our global ensemble is not designed to study this effect. Therefore, we wish to avoid repeating this discussion in the Conclusions section as we feel it would put too much emphasize on it.

---

## Author Comment (AC2) · 19 Sep 2016

**Author's reply to Anonymous Referee #2**

We would like to thank referee #2 for his/her comments that helped improve the quality of the paper. Below we addressed the comments one-by-one; comments from the referee are typeset in italic, our replies are in normal font, and our changes in manuscript are in blue. Line, page and figure numbers in the referee's comments refer to the original manuscript, whereas in our reply we give page and line numbers that refer to the revised manuscript.

**General Comments**

*1. As the retrieval algorithm is firstly introduced in this paper, the authors need to explain the detailed description of the retrieval algorithm. For example, in Table 1, the authors should summarize the detailed a priori information, its structure and units as well as the state vector elements used in the algorithm.*
Following the reviewer's suggestion, we have added the units and a priori information in Table 1. More detailed information on the algorithm is given by Butz et al., 2012 as stated in the text in section 2.1.

*2. The authors should highlight the advantage of the retrieval algorithm, as compared with other algorithms for CH4 monitoring.*
A discussion on the various CH4 retrieval approaches is given in the introduction. Also, mentioned there is that the proxy approach cannot be applied to TROPOMI because of the spectral range. The main advantage of our method compared to other physics-based methods is the computational speed which is an essential performance aspect for the operational data processing of TROPOMI measurements. We have added the following sentence on page 5, line 123:
To our knowledge, our algorithm is among the fastest in use for CH4 full-physics retrievals, with an average CPU time of ~7-0 seconds per retrieval (Hasekamp et al. 2016).

*3. As described in Section 2.1.1, the aerosol type in this retrieval algorithm is characterized by the refractive index and size distribution, which are assumed and fixed parameters in the algorithm. However, aerosols information is the one of the most important factors on XCH4 retrievals and thus the authors should consider the effects of assumed aerosol information on XCH4 retrievals.*
The refractive index is indeed assumed fixed, but this has no great influence on the retrievals as shown by Butz et al 2010. The size distribution is not fixed but retrieved through the size parameter alpha as described in Section 2.1.2. We have added the following sentence on page 6, line 154:
Butz et al. (2010) found that the exact choice of the fixed-value parameters, such as the refractive indices or the width of the height distribution, does not affect the retrievals significantly.

*4. There is unclear description on the following point, A. In Section 2.3, authors describe the filtering criteria to remove the retrievals with bad quality. B. In line 233, authors applied the filtering criteria for the ratio of retrieved H2O column between at weak and strong band. Authors should describe the reason why the ratio is lower than 0.08.*
The physical reason is that the water column retrieved from the strong absorption band is more affected by clouds than the water column retrieved from the weak absorption band as described by Taylor et al 2016 and Frankenberg et 2014. We have added references and explanation on page 10, line 243:
Here we make use of the fact that clouds and aerosols will modify the optical path length in the two bands differently due to the different absorption strengths (Taylor et al., 2016; Frankenberg, 2014). Thus there will be a difference between the H2O column retrieved from the strong band and the H2O column retrieved from

the weak band in the presence of clouds.

The exact value of the threshold has been found by varying the threshold and looking at the quality of the retrievals that pass the filter. Also, the same threshold was found by Scheepmaker et al 2016. We have rewritten the paragraph to clarify this, page 10, line 251:

Because the weak-band and the strong-band retrievals are differently affected by scattering, the ratio |H2O_weak – H2O_strong|/H2O_strong is strongly correlated with cloud contamination. Scenes in which this ratio exceeds a certain threshold can be flagged as cloudy and filtered out. Based on a realistic ensemble of synthetic measurements (see Sect. 4), we find that a threshold of 0.08 filters out most cloudy scenes and keeps most of the clear-sky scenes. However, a direct cloud filter based on VIIRS data is shown to be superior to this indirect two-band cloud approach. Therefore, the two-band cloud filter will only be used as backup when VIIRS data is unavailable.

*5. In Section 3, the authors described the sensitivity of XCH4 retrievals to atmospheric input data and instrument errors. The authors would better describe about the sensitivity to assumed information in the algorithm. In addition, the authors would better show the sensitivity to aerosol information, such as aerosol column, aerosol size parameter and aerosol height parameter.*

We would like to note that the XCH4 error caused by the assumptions in the aerosol parameterization is extensively discussed is the paper. Section 3.1 is almost exclusively dedicated to this topic. Other important a priori assumptions in the algorithm related to temperature, pressure, and vertical profiles of H2O and CH4 are also extensively discussed. In addition, instrument related error sources also form a very important contribution to the total error budget. So, we do not see here the necessity to adopt the manuscript.

*6. In Section 4, cloud information derived from the MODIS products, such as cloud fraction, cloud optical thickness and cloud top height. The authors should briefly explain the method to obtain the cloud information.*

We have added a paragraph on page 21, line 470:

Cloud fraction, cloud optical thickness and cloud top pressure are obtained from MODIS Aqua measurements at 5 x 5 km^2. Here, the cloud top height is derived from the cloud top pressure and the surface pressure, also provided by MODIS. The cloud properties are collocated in time and space to the TROPOMI orbit and used in the measurement simulation. For fractional clouds, the independent-pixel approximation is used to combine the cloudy and clear-sky parts of the scene

*7. In Section 2.3, the authors described the method to filter out scenes including cloud screening. Moreover, in Section 4, the authors also described about cloud screening method and its performance in terms of comparison of two filtering method. The methods are mixed in these sections and is unclear. Please make sure the filtering criteria for post-screening and cloud screening in individual section if you want to simultaneously show its performance.*

Section 2 describe the method while section 4 applies it to synthetic measurements. Following the reviewer's suggestion, we have put the description of a prori filtering and a posteriori filtering in subsections 2.3.1 and 2.3.2, respectively.

**Minor Comments**

*1. In Fig. 3, please add a legend label.*

We have added a legend and updated the caption:

The red shaded area represents the weak absorption band, while the blue area represents the strong absorption band of H2O.

*2. In Fig. 4, is it monthly mean value of retrievals? Please show the period that you*

*have these results. Also, you can plot again with large legend.*
It is explained in the text that we have simulations of one day per season, so it's not monthly mean. To clarify this, we added in the caption of Fig. 4:
The ensemble covers scenes for a day in January (JAN), April (APR), July (JUL), and October (OCT) as described in Sect. 3.1.

*3. Table 2 shows the summarized the effect of instrument calibration errors on the XCH4 bias and precision. The authors would better summarize the effect of atmospheric input data and assumed scattering simultaneously.*
The effect of the scattering forward model error is given in the Table as this is the baseline (first row). To clarify this, we added in the caption:
Note that all sensitivities include the baseline forward model error, caused mainly by aerosol and cirrus scattering.

For the atmospheric input data, we deliberately chose to represent these in a different way (in Fig. 6) as we studied many perturbations and we are convinced that this study would be more difficult to interpret from a tabular form. Therefore, we made no changes here.

*4. In Fig. 9, authors describe the panel (c) and (d) give the valid retrievals after cloud filtering with MODIS and the backup cloud filters. The authors should revised that panel give the XCH4 biases of the valid retrievals after filtering.*
This has been corrected:
Panel (c) and (d) give XCH4 bias of the valid retrievals after cloud filtering with…

*5. In 471, please check spelling in the sentence*
We rephrased the sentence, page 24, line 515:
To achieve this, it is needed to apply a priori filtering of cloud contaminated scenes and a posteriori filtering based on retrieved parameters